# Immuno-Electron and Confocal Laser Scanning Microscopy of the Glycocalyx

**DOI:** 10.3390/biology10050402

**Published:** 2021-05-04

**Authors:** Shailey Gale Twamley, Anke Stach, Heike Heilmann, Berit Söhl-Kielczynski, Verena Stangl, Antje Ludwig, Agnieszka Münster-Wandowski

**Affiliations:** 1Medizinische Klinik für Kardiologie und Angiologie, Charité—Universitätsmedizin Berlin, Corporate Member of Freie Universität Berlin, Humboldt-Universität zu Berlin, and Berlin Institute of Health, 10117 Berlin, Germany; shailey.twamley@charite.de (S.G.T.); anke.stach@charite.de (A.S.); verena.stangl@charite.de (V.S.); 2DZHK (German Centre for Cardiovascular Research), Partner Site, 10117 Berlin, Germany; 3Institute of Integrative Neuroanatomy, Charité—Universitätsmedizin Berlin, Corporate Member of Freie Universität Berlin, Humboldt-Universität zu Berlin, and Berlin Institute of Health, 10117 Berlin, Germany; heike.heilmann@charite.de; 4Institute for Integrative Neurophysiology—Universitätsmedizin Berlin, Corporate Member of Freie Universität Berlin, Humboldt-Universität zu Berlin, and Berlin Institute of Health, 10117 Berlin, Germany; berit.soehl-kielczynski@charite.de

**Keywords:** glycocalyx, immuno-electron microscopy, high-pressure freezing, freeze-substitution, pre-embedding immunogold labeling, confocal laser scanning microscopy

## Abstract

**Simple Summary:**

The glycocalyx (GCX) is a hydrated, gel-like layer of biological macromolecules attached to the cell membrane. The GCX acts as a barrier and regulates the entry of external substances into the cell. The function of the GCX is highly dependent on its structure and composition. Pathogenic factors can affect the protective structure of the GCX. We know very little about the three-dimensional organization of the GXC. The tiny and delicate structures of the GCX are difficult to study by microscopic techniques. In this study, we evaluated a method to preserve and label sensitive GCX components with antibodies for high-resolution microscopy analysis. High-resolution microscopy is a powerful tool because it allows visualization of ultra-small components and biological interactions. Our method can be used as a tool to better understand the role of the GCX during the development and progression of diseases, such as viral infections, tumor invasion, and the development of atherosclerosis.

**Abstract:**

The glycocalyx (GCX), a pericellular carbohydrate rich hydrogel, forms a selective barrier that shields the cellular membrane, provides mechanical support, and regulates the transport and diffusion of molecules. The GCX is a fragile structure, making it difficult to study by transmission electron microscopy (TEM) and confocal laser scanning microscopy (CLSM). Sample preparation by conventional chemical fixation destroys the GCX, giving a false impression of its organization. An additional challenge is to process the GCX in a way that preserves its morphology and enhanced antigenicity to study its cell-specific composition. The aim of this study was to provide a protocol to preserve both antigen accessibility and the unique morphology of the GCX. We established a combined high pressure freezing (HPF), osmium-free freeze substitution (FS), rehydration, and pre-embedding immunogold labeling method for TEM. Our results showed specific immunogold labeling of GCX components expressed in human monocytic THP-1 cells, hyaluronic acid receptor (CD44) and chondroitin sulfate (CS), and maintained a well-preserved GCX morphology. We adapted the protocol for antigen localization by CLSM and confirmed the specific distribution pattern of GCX components. The presented combination of HPF, FS, rehydration, and immunolabeling for both TEM and CLSM offers the possibility for analyzing the morphology and composition of the unique GCX structure.

## 1. Introduction

The glycocalyx (GCX), a carbohydrate rich hydrogel located on the surface of cells, provides mechanical support and forms the selective barrier between the cell surface and the extracellular space [1]. It is involved in many processes that are related to the cell membrane such as blastocyst implantation, embryonic development, leukocyte adhesion, and viral and bacterial infections [2,3,4]. Glycoproteins (GPs) and proteoglycans (PGs), together with covalently bound glycosaminoglycan (GAG) chains and sialic acid, are the backbone molecules of the GCX. Extending far into the extracellular space, GAG chains such as heparan sulfate (HS), chondroitin sulfate (CS), and hyaluronic acid (HA) are linked and/or intertwined with specific PGs in combination with other ECM proteins [4]. The outer surface layer is more loosely associated with proteins, growth factors, ions, and other biological components [5]. The detailed composition and structure of the GCX varies according to the cell type and determines its functionality under physiological and pathophysiological conditions [4,6]. However, little is known about the structure and dynamics of the GCX in different cell types on an ultrastructural level. This knowledge gap is largely due to a lack of methods that allow both structural preservation and immunolabeling of GCX-specific components.

The GCX is a fragile structure, making it difficult to study by TEM and CLSM. Sample preparation by conventional aldehyde fixation followed by alcohol dehydration causes the GCX to collapse, resulting in an incorrect estimation of its dimension and organization [7]. It was established that rapid cryofixation followed by freeze substitution (FS) is a more suitable method for preserving the ultrastructure of the GCX for TEM [8,9]. Ebong and co-workers showed that after rapid freezing (RF) and subsequent FS, the GCX of cultured bovine endothelial cells in the TEM appeared as a several µm thick mesh-like structure, which was clearly different to the collapsed state after chemical fixation [8]. Recently, Poller et al. (2020) used a combined high pressure freezing (HPF) and FS method on human monocytic THP-1 cells, resulting in an equally well-preserved ultrastructure of the GCX [9].

Immuno-electron microscopy is a powerful technique for observing the cellular, subcellular, and extracellular localization of antigens and for studying the relationship between antigens and other macromolecules. The rapid cryofixation established for TEM-suitable GCX preservation can be applied before pre- and post-embedding immunolabelling methods [10,11]. In order to stabilize the fragile GCX during resin embedding and polymerization after rapid cryofixation, the FS media used in the studies of Ebong and Poller contained osmium tetroxide (OsO_4_) [8,9]. However, since OsO_4_ cross-links proteins and polypeptide chains and thus affects antigen–antibody reactions [12], incubation with osmium is generally not recommended prior to immunoreactions.

The aim of this study was to provide a method for electron and confocal microscopic analysis of the GCX that warrants both antigen accessibility and morphological preservation. We present a combined HPF, FS, rehydration, and pre-embedding immunogold labeling protocol that enables high quality immunocytochemistry while maintaining GCX morphology at the ultrastructural level.

## 2. Materials and Methods

### 2.1. Cell Culture

To establish our HPF/FS/rehydration and pre-embedding immunolabeling method we used THP-1 cells. THP-1 cells (human acute monocytic leukemia cell line; ATCC, Wesel, Germany) were cultured in suspension in a humidified incubator at 37 °C with 5% CO_2_ in RPMI-1640 medium (Invitrogen, Karlsruhe, Germany), supplemented with 10% fetal calf serum (FCS; Biochrom, Berlin, Germany), 100 U/mL penicillin, 100 μg/mL streptomycin (Invitrogen), and 2 mM L-glutamine (Invitrogen). Exact cell numbers were determined with a hemocytometer.

### 2.2. Formalin Fixation

THP-1 cells were washed with phosphate buffered saline (PBS, Thermo Fisher, Waltham, MA, USA) and fixed in phosphate buffered 4% formalin solution (Roti-Histofix, Carl Roth GmbH, Karlsruhe, Germany) for 10 min at room temperature (RT). Samples were washed with PBS and blocked for 1 h at room temperature in PBS with 5% normal goat serum (NGS) and 2% bovine serum albumin (BSA) (blocking buffer). Primary antibodies (Table 1, CD44 (IM7) FITC, CS-56) were incubated overnight at 4 °C in the dark in blocking buffer. Next, samples that were incubated with primary antibodies for CS were washed and incubated with secondary antibody (goat anti-mouse IgM Alexa Fluor-488; Invitrogen, Karlsruhe, Germany) suspended in PBS with 2% NGS overnight at 4 °C. After immunofluorescence labeling was complete, cells were stained for 5 min in the dark with 4′,6-diamidin-2-phenylindol (DAPI) (1 µg/mL). After washing and resuspension in PBS, cells were added to a 4-well µ-slide (ibidi GmbH, Gräfelfing, Germany) and allowed to settle on the bottom for 15 min. First, immunolabeled THP-1 cells were imaged for CS-56 and FITC CD44-IM7 labeling. Following immunofluorescence, Texas red-conjugated WGA (1 μg/mL; Molecular Probes, Eugene, OR, USA) was added to each well, while cells were still in focus on the microscope stage, in a drop-by-drop manner and imaged immediately. Cells were imaged with a Nikon scanning confocal A1Rsi+ microscope at 60× resolution in a water immersion.

### 2.3. High Pressure Freezing

In order to obtain a well-preserved GCX and its antigenicity we applied HPF as a fixation method. THP-1 cells, in a concentration of 10^6^/mL were sedimented by gravity in 1.5 mL tubes. An aliquot of the cell sediment was transferred into gold carriers (6 mm diameter, 200 µm indentation; Leica, Wetzlar, Germany), previously coated with 1-hexadecene (non-penetrating cryoprotectant [13]; Sigma-Aldrich, Germany), and immediately frozen in an EM HPM 100 High Pressure Freezer (Leica, Wetzlar, Germany) after reaching 2100 bar under liquid nitrogen. The detailed process of high-pressure freezing using the HPM 100 can be found in Studer et al. (2008) [10]. The frozen carriers were collected in a liquid nitrogen filled chamber and transferred to the freeze substitution unit (EM-AFS; Leica Microsystems, Wetzlar, Germany), previously cooled to −90 °C.

### 2.4. Freeze Substitution and Rehydration

For TEM, frozen samples were slowly freeze substituted for 24–48 h at −90 °C in FS-cocktail containing 0.2% uranyl acetate (UAc; Merck, Germany), 0.2% glutaraldehyde (GA; Electron Microscopy Sciences, Hatfield, PA, USA), 1% methanol (Roche, Mannheim, Germany), and 5% water in acetone (Roche, Mannheim, Germany). For CLSM, we removed UAc from the FS-substitution cocktail since this toxic and radioactive reagent is not necessary for this purpose. The temperature was gradually increased (5 °C/1 h) to 0 °C. Samples were then transferred to a cooled chamber. Afterwards the sample holders were carefully removed, and cells were gently centrifuged (100 rpm, 15 s at 4 °C), washed twice with pure acetone (Roche, Mannheim, Germany), and centrifuged again. Next, cells were rehydrated in increasing concentrations of water/phosphate buffer in acetone and were processed for immunolabeling for TEM and CLSM. Please also refer to the figures describing the workflows in the results section.

### 2.5. Pre-Embedding Immunogold Labeling and Processing for TEM

In order to analyze the distribution of the cell surface receptor for hyaluronic acid (CD44) and chondroitin sulfate (CS56) in the GCX of THP1-cells, we applied pre-embedding immunogold labeling for TEM analysis. For a list of the primary antibodies, refer to Table 1. Rehydrated cells were washed twice with phosphate buffer (PB). To avoid nonspecific binding, samples were incubated for 30 min in blocking solution containing PB, 5% NGS (PAN Biotech), and 2% BSA (Sigma-Aldrich, Darmstadt, Germany). All of the following immuno-incubations were done with gentle agitation, overnight at 4 °C. After each incubation step, cells were gently centrifuged. After blocking, suspension cells were incubated with primary antibodies: CD44 and CS, both diluted in blocking solution. After washing with PB, cells were incubated with secondary gold conjugated antibodies: goat anti-rat IgG 30 nm Gold (BBInternational, Cardiff, UK) and goat anti-mouse IgM 25 nm Gold (Aurion, Wageningen, The Netherlands), both diluted 1:20 in PB containing 0.2% acetylated BSA (BSA-c.; Aurion, Wageningen, The Netherlands). To remove unbound secondary antibodies, cells were washed thoroughly with PB/BSA-c and then with PBS. Then, samples were post-fixed with 2% GA in PB for 30 min to crosslink gold conjugates in the structures, which prevents the loss of labeling during subsequent processing. Next, cells were washed several times in PB and incubated additional 30 min with buffered 0.5% OsO_4_ for structural stabilization. Next, cells were washed in PB and in ddH_2_O, dehydrated in increasing concentrations of ethanol and finally embedded in epoxy embedding medium (Epon 812; Sigma-Aldrich, Darmstadt, Germany). After resin polymerization at 60 °C, cells were ultra-sectioned into 60–65 nm on an ultramicrotome (Reichert Ultracut S, Leica, Germany) with a diamond knife (Diatome, Nidau, Switzerland) and mounted on 300-mesh Formvar-coated nickel grids (Plano, Wetzlar, Germany). Ultrathin sections were examined, without further staining, using a Zeiss transmission electron microscope 912 (TEM-912, Carl Zeiss, Oberkochen, Germany) operating at 120 kV and equipped with a digital camera (Proscan 2K Slow-Scan CCD-Camera, Carl Zeiss, Oberkochen, Germany). Digital image acquisition was performed using the ImageSP software (TRS, Moorenweis, Germany).

### 2.6. Immunofluorescence Labeling and Confocal Imaging

Rehydrated cells were incubated for 15 min in blocking buffer containing PB, 5% NGS (PAN Biotech), and 2% BSA (Sigma-Aldrich, Darmstadt, Germany). Primary antibodies (Table 1, CD44 (IM7), CS-56) were incubated overnight at 4 °C in the dark in blocking buffer. Next, samples were washed and incubated with secondary antibodies (anti-rat IgG Alexa Fluor-488 and anti-mouse IgM Alexa Fluor-488 (Invitrogen, Karlsruhe, Germany)) and suspended in PBS with 2% NGS overnight at 4 °C. Nuclei were stained by a 5 min incubation with a DAPI solution (Sigma, St. Louis, MO, USA; 100 ng/mL). Labeled cells were resuspended in PB and transferred to µ-slides I 0.4 Luer (Ibidi, Gräfelfing, Germany) before imaging with a laser scanning confocal microscope (FV1000, Olympus, Tokyo, Japan).

We imaged cells first at low magnification (20×, Olympus, Tokyo, Japan) and arranged the overview images. Higher resolution image stacks were acquired using a 60× silicon oil immersion lens (0.5 μm step size, a numerical aperture of 1.30; UPlanSApo, Olympus, Tokyo, Japan) with a zoom factor of either 1 or 4 to resolve precise cellular and extracellular localization. Co-staining with Texas red-conjugated WGA (1 μg/mL; Molecular Probes, Eugene, OR, USA) was used for visualization of the GCX. WGA was added directly to the µ-slides just before imaging. The images were acquired through separate channels and temporally non-overlapping excitations of the fluorochromes, and they were analyzed off-line using the ImageJ software package (courtesy of W.S. Rasband, U.S. National Institutes of Health, Bethesda, Maryland, http://rsb.info.nih.gov/ij/, accessed on 14 December 2020).

### 2.7. Control Experiments

As a negative control for immunolabeling, primary antibodies against CD44 and CS were omitted. To exclude the possibility that the pericellular mesh-like structures were caused by the formation of medium-derived artifacts during HPF/FS, we fixed cell culture medium containing 10% FCS in the same manner as THP-1 cells and processed it for TEM.

## 3. Results

### 3.1. Antigen Presentation after Formalin Fixation of THP-1 Cells

We used the human myelomonocytic cell line THP-1, a model for investigating monocyte structure and function [14]. THP-1 cells grow as a suspension culture, which enables the investigation of their native pericellular environment without the disturbing effects of, e.g., trypsinization. It has been previously shown that THP-1 cells express the membrane-bound cluster of differentiation 44 (CD44) glycoprotein antigen [15,16]. Chondroitin sulfate (CS) is the main glycosaminoglycan synthesized by THP-1 [16,17].

Figure 1 shows the distribution of CD44 and CS in THP-1 cells after mild chemical fixation with 4% formaldehyde detected with CLSM. Labeling with CD44 IM7 and CS-56 antibodies resulted in signals mainly along the plasma membrane (Figure 1A,B). The signal distribution of CD44 shows uniform labeling while CS appears as spots scattered along the plasma membrane. In the negative controls, without primary antibody incubation, no signal was detected (Figure 1C). The staining with Texas red-conjugated WGA, a lectin that preferentially binds to cell membrane associated N-acetyl-D-glucosamine and sialic acid [18], resulted in a labelling of the plasma membrane (Figure 1D).

Taken together, after mild chemical fixation, CD44, CS and WGA-stain appeared to be strictly membrane associated. Signals for CD44, CS, and WGA were not detectable in the pericellular surrounding.

### 3.2. A Combined HPF, FS, Rehydration, and Pre-Embedding Immunogold Method for Preserved Ultrastructure and Antigen Localization in the GCX of THP-1 Cells by TEM

The starting point for our method evaluation was the earlier published techniques for preservation of the ultrastructure of the GCX using of RF or HPF and FS with osmium tetroxide as the primary fixative [8,9]. We first tested an FS method avoiding osmium and using generally low concentrations of fixative to ensure both good preservation and antigenicity of the GCX. We tested several cocktails for FS, containing different concentrations of uranyl acetate (UAc) and glutaraldehyde (GA). The best results were obtained with a FS-cocktail containing 0.2% UAc, 0.2% GA, 1% methanol, and 5% water in acetone (for method overview see Figure 2).

Figure 3 shows the morphological preservation of the GCX using an FS cocktail with osmium (A, B) and with the modified osmium-free FS cocktail (C, D). Both conditions display no deviation in the preservation quality and the thickness of the GCX (Table 2).

The GCX appears as a pericellular mesh-like structure with an extension of several micrometers (Figure 3A–D). While lipid-retaining osmium treatment resulted in a strong contrast of intracellular membranes (Figure 3A,B), aldehyde/UAc treatment resulted in a lower and more diffuse contrast (Figure 3C,D). In general, intracellular membranes were not strongly contrasted by UAc. Some intracellular structures such as mitochondria appeared as negative images after aldehyde/UAc treatment. We observed no freeze-induced damage to THP-1 cells. Likewise, the rehydration procedure required before immunolabeling had no adverse effects on GCX morphology (Figure 3E,F).

Next, we applied the optimized FS protocol for immunogold pre-embedding labeling using CD44 IM7 antibody against cell surface receptors for hyaluronic acid and CS-56 antibody against chondroitin sulfate. We particularly focused on immunogold detection within the GCX rather than particle quantification. The distribution of the electron dense gold particles showed that CD44 was detectable along the cell membrane and in the more distant areas of the GCX (Figure 4A–C). Immunogold labeling for CS was not associated with the plasma membrane but scattered within the GCX (Figure 4D–F). In both cases, gold particles were not observed intracellularly, which was expected, since samples were not permeabilized. We verified the specificity of immunogold labeling for TEM (see Section 2.4).

Taken together, the optimized combined HPF/FS, rehydration, and pre-embedding immunogold labeling protocol enabled high immunoreactivity while preserving the GCX.

### 3.3. Adjustment of the Method for Confocal Laser Scanning Microscopy

We optimized the method for the application in CLSM (for method overview see Figure 5). We fixed the THP-1 cells with HPF using the same conditions as those for TEM. For FS we removed UAc and instead we used a medium containing only 0.2% GA, 1% methanol, and 5% water in acetone. After FS and rehydration, THP-1 cells were labeled for CD44 and CS (Figure 6A,B). In addition, we performed WGA staining to visualize sugar components of the GCX. DAPI was used as a nuclear stain (Figure 6C,D).

The most noticeable difference compared to the labeling after conventional chemical fixation was the shift from a strict membrane labeling by WGA to a diffuse signal extending into the pericellular environment (Figure 6C,D). In contrast to the appearance of CD44 labeling in aldehyde-fixed samples, the CD44 signal did not appear as a sharp cell border, it extended into the pericellular space (Figure 6A,C), similarly to the distribution pattern observed by TEM. Moreover, the dispersed distribution of CD44 near the plasma membrane and within the GCX overlapped with the WGA-stained areas (Figure 6C). Immunofluorescence labeling for CS showed, corresponding to the TEM results, clusters of different sizes (Figure 6B,D). Overlap of CS with the WGA stain indicated localization in the GCX (Figure 6D). Thus, consistent with pre-embedding immunogold labeling for CD44 and CS in TEM, CLSM revealed comparable distribution within the GCX.

### 3.4. Control Experiments

We carried out a series of control experiments for TEM and CLSM to confirm the specificity of the immunolabeling. At the ultrastructural level, control sections with omitted primary antibody incubation did not reveal any intra- and extracellular specific labeling (Figure 7A). Occasionally, some single gold particles were observed extracellularly but did not show association with GCX-like structures (Figure 7A, arrow). Additionally, we analyzed cell culture media containing 10% FCS that was processed for TEM and embedded in the same manner as that used for the THP-1 cells. We did not observe any mesh-like arrangements of medium components (Figure 7B). This allowed us to rule out that the pericellular mesh-like structures are artifacts formed from proteins in the cell culture medium. The conjugated secondary antibodies used for CLSM resulted in a weak, uniform, non-specific background, observed in the nuclei and cytoplasm of THP-1 cells (Figure 7C,D).

## 4. Discussion

We introduced a method for microscopic analysis of the GCX that warrants both antigen accessibility and morphological preservation. We present a combined HPF/FS, rehydration, and pre-embedding immunogold labeling protocol for TEM that enables high quality immunocytochemistry while maintaining GCX morphology at the ultrastructural level. We adapted the protocol for application in immunofluorescence labeling and imaging by CLSM.

The investigation of the GCX with electron and confocal microscopy is challenging. The most critical step for both microscopic techniques is the fixation of the highly hydrated GCX while maintaining its full volume. Conventional chemical fixation leads to the collapse of fragile extracellular components and therefore does not show the actual dimension of the GCX. Specific chemical fixations using ruthenium red in combination with OsO_4_ or lanthanum nitrate with alcian blue only partly improved the preservation of the GCX [19,20]. Because measurements of endothelial GCX thickness in vivo showed significantly larger dimensions than could be observed on cultured cells by TEM, it was suggested that the GCX has an impressive expansion in vivo but not in vitro [21]. This view was challenged by Ebong et al. (2011), who demonstrated that the GCX of in vitro cultivated endothelial cells can be preserved by rapid slam freezing (RF) combined with freeze substitution [8]. RF stabilizes biological samples to an amorphous hydrated state, which maintains the hydrogel structure of the GCX much better than does chemical fixation [8]. In the present work, we used HPF, which, in contrast to RF, uses extremely high pressures (2100 bar) in combination with low temperature (−180 °C), which allows for an instantaneous conversion of liquid water into amorphous ice, thereby reducing the formation of ice crystals. It was shown that HPF immobilizes cellular structures in their near-native form within milliseconds and that subsequent dehydration and contrasting by FS allows for the preservation of ultrastructure and antigenicity due to a gradual temperature rise that successively replaces amorphous ice with organic solvents [10,22]. HPF/FS has recently been successfully applied to visualize the interaction of magnetic nanoparticles with the GCX of THP-1 cells during initial contact [9] and the extracellular surface layer of visceral organs [23].

The fundamental study by Ebong et al. (2011) highlighted the need for the presence of albumin during rapid freezing for the preservation of the GCX [8]. The barrier function of the GCX depends on its interaction with plasma-derived proteins, in particular albumin [24]. Loss of circulating albumin accompanies loss of the GCX and even low concentrations support GCX stability in vivo [25,26,27]. The logical consequence is that the structure of the GCX can only be preserved by maintaining an optimal protein content during cryofixation. Instead of buffer with albumin, we used cell culture medium with 10% FCS for HPF. FCS contains ~7% protein; more than a half of it is albumin. This allowed us to concentrate the cells by sedimentation in the original medium without media changes and additional centrifugation. In this way, the cells were exposed to minimal mechanical manipulation and did not have to change their milieu before freezing. We assume that the pericellular structure obtained by this method reflects as closely as possible the actual GCX dimension of cultivated THP-1 cells.

The FS medium used in the previous TEM studies of the GCX in cultured endothelial and monocytic cells contained osmium tetroxide (OsO_4_) [8,9], which was considered necessary for stability when embedded in plastic resins such as Epon and for polymerization at higher temperatures [28]. It is known that OsO_4_ as a primary fixative inhibits antigen–antibody reactions, since OsO_4_ cleaves polypeptides in tryptophan residues and oxidizes methionine to methionine sulfone and cysteine to cysteic acid [12]. To obtain reliable immunolabeling results, we avoided the substitution with osmium. Instead, we added GA and UAc in low concentrations and water (5%, *v/v*) to the FS medium. UAc is known to increase the visibility of structures, such as actin filaments and nucleic acids [29,30] and to preserve antigenicity better than osmium. The addition of water to FS medium was reported to improve the visibility of membrane structures [31]. We found that the thickness and appearance of the GCX was not affected by the absence of OsO_4_ during FS, suggesting that OsO_4_ during the substitution is not essential for the structural integrity of the GCX. Omission of osmium in the FS medium resulted in lower contrast of membranes and intracellular structures, but the general morphology of the cell was well visible.

An important step toward consistent immunolabeling results was the optimization of the substitution timing, with the aim of avoiding the formation of even small ice crystals that could potentially penetrate cellular and extracellular structures and thus affect preservation and immunoreactivity. The slow temperature rise of 5 °C/hour applied in our protocol was crucial to preserve cells with marginal or no damage by freezing or freeze-substitution. Indeed, we observed no freeze-induced damage to THP-1 cells.

For TEM analysis of the samples, we decided to perform pre-embedding immunogold labeling to avoid masking of the antigens by resin [11,32]. Because visualization of antigens in the GCX did not require immunoreagents to penetrate the plasma membrane, we omitted permeabilization steps. The rehydration procedure prior to antibody labeling [33] enabled the subsequent aqueous reactions with primary- and secondary antibodies after FS, as proven by the successful detection of CD44 and CS in the GCX of THP-1 cells. Because THP-1 cells mainly express Versican, a CS proteoglycan, and because CS is the most abundant GAG as demonstrated by HPLC analysis [17], we assumed that CS is a major component of the GCX of THP-1 cells. In fact, after HPF/FS we found a clear pericellular signal with the CS-56 antibody, which was described as recognizing an unspecified sugar motif of CS-PG [34]. We observed a cluster-like appearance of staining with the CS-56 antibody as previously described by others [34]. The adhesion molecule CD44, a CS proteoglycan and hyaluronic acid (HA) receptor, has long been viewed as a transmembrane protein [35]. The distribution of CD44 on THP-1 cells after formalin fixation indeed gives the impression that CD44 is exclusively a membrane molecule. However, after fixation by HPF/FS we could detect membrane-associated CD44 along the cell surface as well as a non-transmembrane form of CD44 dispersed in the GCX. This observation supports the notion that CD44 also occurs as a soluble component (sCD44) in the pericellular matrix of cells [35,36].

It was important for us to test whether our protocol could be adapted to study the GCX in a near native state using CLSM. Compared to TEM, CLSM provides a faster overview of larger regions of interest. In addition, TEM is a time-consuming and expensive technique that is not available in every laboratory. With slight modifications in the FS procedure, we translated our protocol for CLSM. The mesh-like structure of the GCX was not resolved in CLSM. However, staining with fluorescently labeled WGA enabled the visualization of the preserved GCX and showed its approximate dimensions. The lectin WGA binds to sialic acid, hyaluronic acid, and CS-A [37] and fluorescently labeled WGA was previously used for direct visualization of the endothelial GCX in vivo [38]. During our experiments, we found that the WGA staining of the GCX after HPF/FS is not stable over time. Within minutes, the structure of the stained region changed from uniform staining of the pericellular area to non-uniform local concentration of staining. This could be due to agglutination, implying that the natural state of the WGA binding partners was remarkably well preserved. Despite fast agglutination, which limited the imaging of z-series, we observed clear overlapping of CD44 and CS with the WGA-stained GCX. Overall, CLSM allowed us to confirm the distribution patterns of CS and CD44 observed with TEM, demonstrating the reliability of the method.

Limitations and future perspectives: In this proof-of-principle study, we used THP-1 cells as an easy-to-handle experimental model. However, the use of suspension cells required several centrifugation steps during immunolabeling, which favored clustering of the cells. We cannot exclude a resulting impact on the structure of the GCX. The application of our protocol to other cell types (e.g., primary leukocytes, adherent cells) or to tissues certainly requires adaptations of the HPF/FS protocol. The use of adherent cells can lead to advantages such as the elimination of centrifugation steps and better accessibility of antibodies. If permeabilization is required, the detergent, its concentration, and incubation conditions must be carefully selected to avoid damage to the cellular [11] and GCX ultrastructure. The presented combination of HPF, FS, rehydration, and pre-embedding immunogold labeling or immunofluorescence labeling offers the possibility for analyzing the morphology and composition of the unique GCX structure. Using this protocol, we were able to reveal distribution patterns of surface-associated biomolecules in the GCX of THP-1 cells with TEM and CLSM. The protocol is potentially applicable to 3D electron tomography [11]. This could enable the study of the three-dimensional architecture of the GCX and the distribution of its components. The method can be used as a tool to visualize pathological changes of the GCX and its barrier function, to study the interaction of viruses and nanoparticles with the GCX, and to investigate cell–cell interactions. This is not only of high interest for immune cell research, but also for the investigation of endothelial and epithelial cell biology.

## 5. Conclusions

The presented combination of HPF, FS, rehydration, and pre-embedding immunogold labeling or immunofluorescence labeling offers the possibility for analyzing the morphology and composition of the unique GCX structure. Using this protocol, we were able to reveal distribution patterns of surface-associated biomolecules in the GCX of THP-1 cells with TEM and CLSM. The protocol is potentially applicable to 3D electron tomography [11]. This could enable the study of the three-dimensional architecture of the GCX and the distribution of its components. The method can be used as a tool to visualize pathological changes of the GCX and its barrier function, to study the interaction of viruses and nanoparticles with the GCX, and to investigate cell–cell interactions. This is not only of high interest for immune cell research, but also for the investigation of endothelial and epithelial cell biology.

## Figures and Tables

**Figure 1 biology-10-00402-f001:**
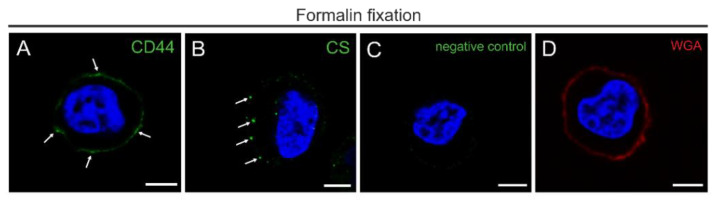
Morphology and antigenicity of the cell surface in THP-1 cells after formalin fixation. (**A**,**B**) CSLM images showing immunofluorescence single labeling for GCX markers: (**A**) CD44 (green pseudocolor) and (**B**) CS (green pseudocolor) with (**C**) the corresponding negative control with omitted primary antibody. Note the CD44-labeling and scattered clusters of CS close to the plasma membrane (arrows). (**D**) WGA staining (red pseudocolor). (**A**–**D**) DAPI fluorescence stains for nuclear DNA (blue pseudocolor) Scale bar: (**A**–**D**) 5 µm.

**Figure 2 biology-10-00402-f002:**
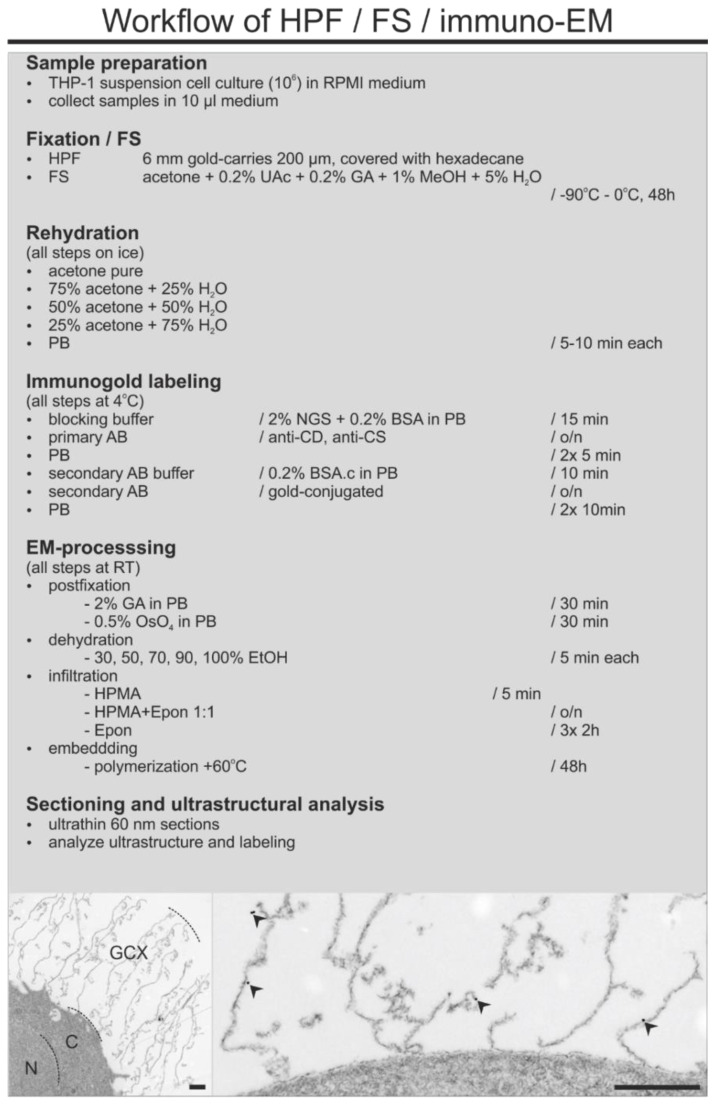
Flowchart showing the workflow of the combined HPF/FS/rehydration and pre-embedding immunogold labeling method for TEM. Insert on the left side shows well-preserved intra- and pericellular ultrastructure of THP-1 cells after processing for immuno-TEM. Note the dimension of the GCX. The dashed lines mark the border of the nucleus to cytoplasm and cytoplasm to GCX. Insert on the right shows a higher magnification of the GCX structures with immunogold labeling for CD44 (arrowheads). N, nucleus; C, cytoplasm; GCX, glycocalyx. Scale bar: 1000 nm.

**Figure 3 biology-10-00402-f003:**
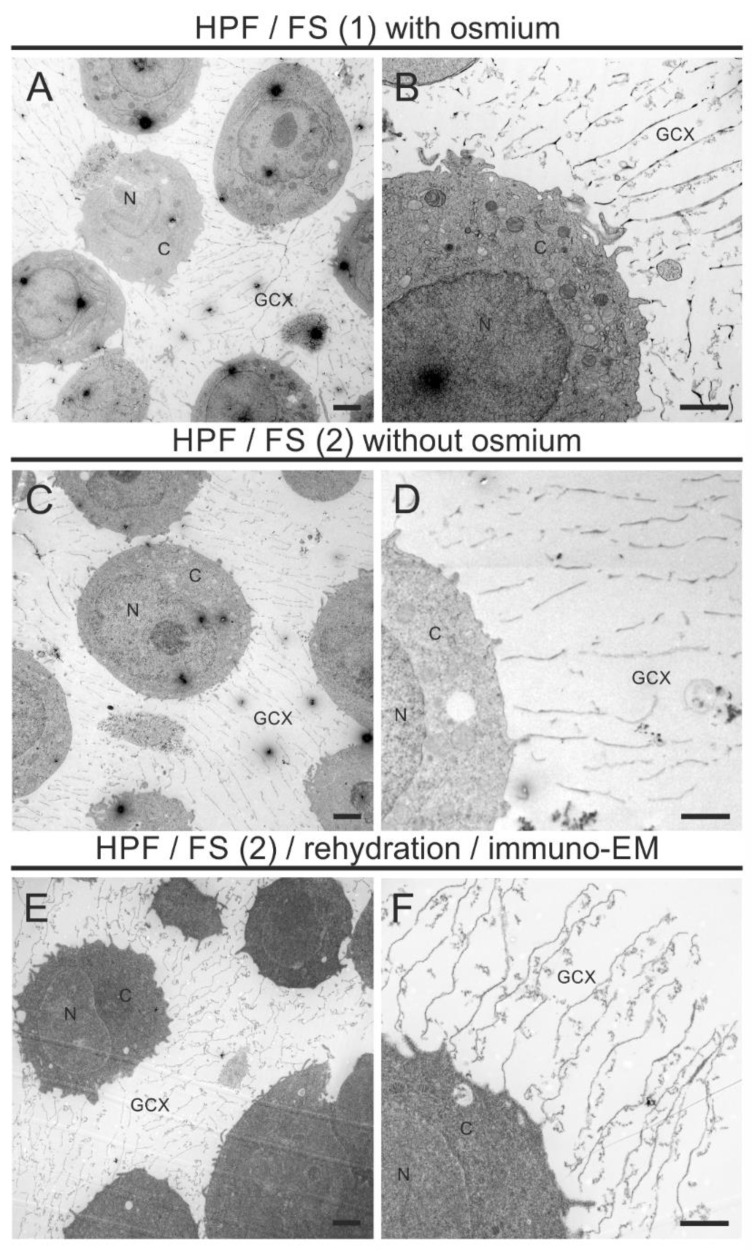
Immuno-TEM images of the ultrastructure of the GCX of THP-1 cells. The electron micrographs show embedded THP-1 cells at lower magnification for a morphological overview of the cellular and extracellular structures (**A**,**C**,**E**) and at higher magnification for a detailed intra- and pericellular morphology of individual THP-1 cells (**B**,**D**,**F**) after standard osmication (**A**,**B**), optimized aldehyde/UAc substitution (**C**,**D**), following rehydration (**E**,**F**). Note the better preservation of cellular membranes after FS with osmium (**A**,**B**) and consistently well-preserved ultrastructure of the GCX after osmication, optimized FS, following rehydration. N, nucleus; C, cytoplasm; GCX, glycocalyx. Scale bar: (**A**,**C**,**E**) 2500 nm; (**B**,**D**,**F**) 1000 nm.

**Figure 4 biology-10-00402-f004:**
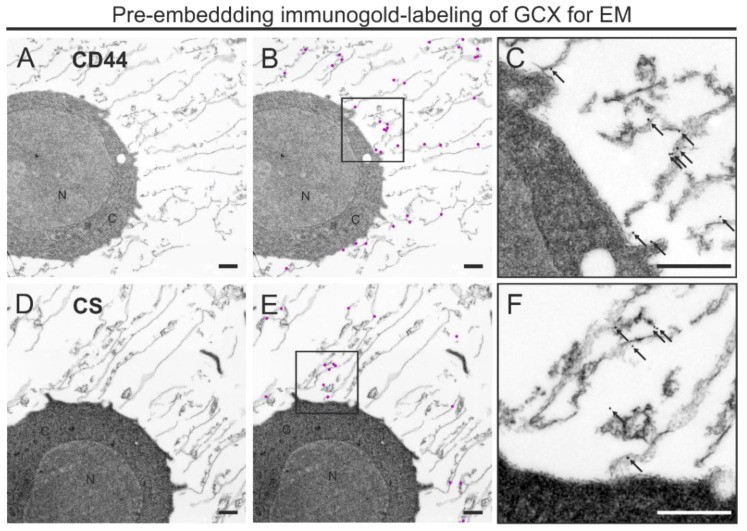
Pre-embedding immunogold labeling for GCX antigens of THP-1 cells after HPF, osmium-free FS, and rehydration. (**A**–**F**) TEM micrographs showing immunogold labeling for CD44 (**A**–**C**) and CS (**D**–**F**) in the GCX of THP-1 cells. (**A**,**D**) GCX structures labeled by antibodies directed against either CD44 (**A**) or CS (**D**), indicated by 30 nm or 25 nm gold particles, respectively. (**B**,**E**) Same images as in (**A**) and in (**D**), with gold particles graphically highlighted in magenta to illustrate the distribution pattern and labeling density of CD44 and CS in the GCX. (**C**,**F**) Magnification of the highlighted region in B and E showing the localization of colloidal gold particles in the original size. N, nucleus; C, cytoplasm. Scale bar: (**A**–**F**) 1000 nm.

**Figure 5 biology-10-00402-f005:**
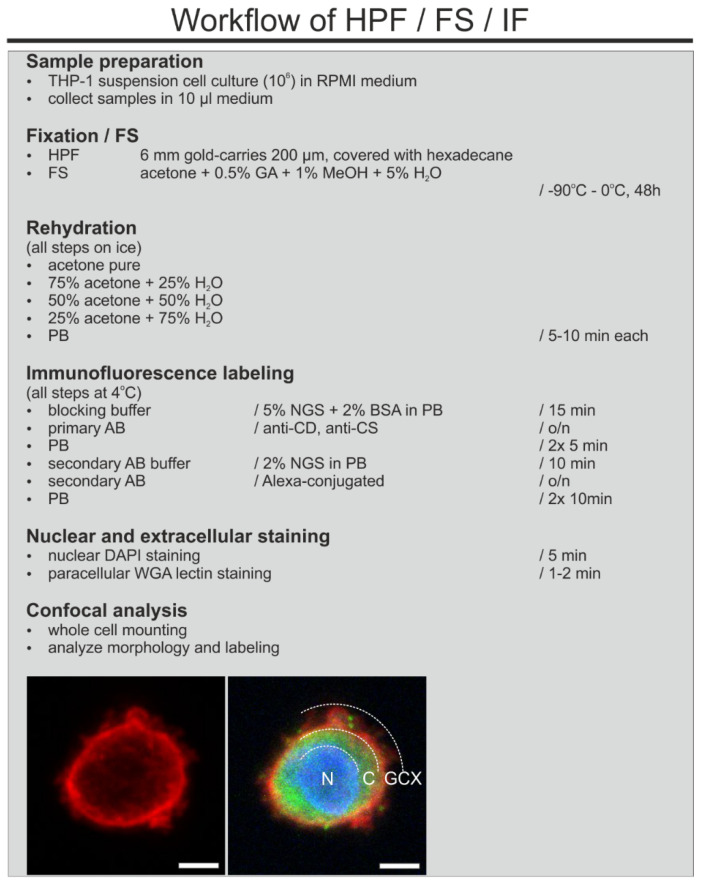
Flowchart showing the workflow of combined HPF/FS/rehydration, and immunofluorescence labeling for CLSM. Insert on the left side shows THP-1 cells with WGA staining that exclusively binds to GCX structures. Note that the WGA staining is not restricted to the plasma membrane but extends to the pericellular region. Insert on the right presents a merged image of immunofluorescence labeling for CS (green pseudocolor), DAPI staining (blue pseudocolor), and WGA staining (red pseudocolor). The dashed lines label the nuclear-to-cytoplasmic and cytoplasmic-to-GCX borders to visualize the extension of GCX into the pericellular region of the THP-1 cell. Scale bar: 5 µm.

**Figure 6 biology-10-00402-f006:**
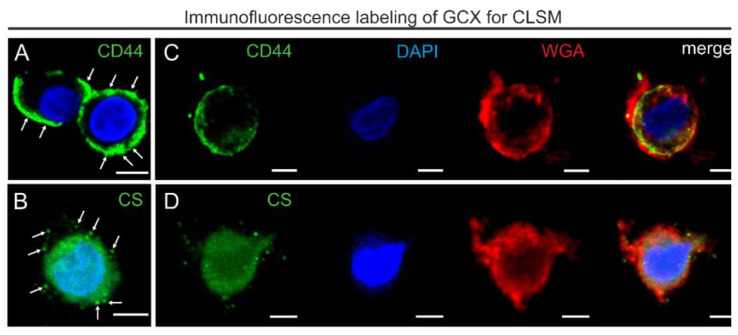
CLSM images of immunofluorescence labeling of CD44 and CS after HPF, FS, and rehydration. (**A**,**B**) CLSM images showing immunofluorescence single labeling for GCX markers: CD44 (**A**; green pseudocolor) and CS (**B**; green pseudocolor). Note that the CD44-labeling (arrows) and scattered clusters of CS (arrows) extend beyond the plasma membrane into the GCX. (**C**,**D**) Additional co-staining for DAPI (blue pseudocolor) and WGA (red pseudocolor). The last images on the right side show colocalization of CD44 or CS immunofluorescence and WGA staining extending into the pericellular region. Scale bar: (**A**–**D**) 5 µm.

**Figure 7 biology-10-00402-f007:**
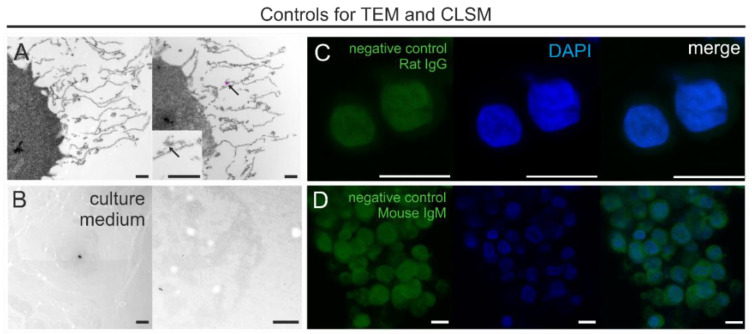
Control experiments. (**A**) Immuno-EM of negative controls with omitted primary antibodies. Note the very rare presence of single gold particles (arrow). (**B**) THP-1 growth medium processed for TEM exhibits no GCX-like structures. (**C**,**D**) CLSM images of negative controls with omitted antibodies for CD44 (**C**) and CS (**D**), co-staining with DAPI, and colocalization of unspecific signals for conjugated secondary antibodies. Note the weak cytoplasmic immunofluorescence background for the secondary antibody for CS (**D**). Scale bars: (**A**,**B**) 1000 nm; (**C**,**D**) 10 µm.

**Table 1 biology-10-00402-t001:** List of primary antibodies.

Antibody	Supplier	Host	Dilution
CD44 (IM7) FITC	Invitrogen 14-0441-85	rat/IgG2b monoclonal	CLSM, 1:500
CD44 (IM7)	Invitrogen 14-0441-82	rat/IgG2b monoclonal	TEM, 1:100 CLSM, 1:500
CS-56	Abcam ab11570	mouse/IgM monoclonal	TEM, 1:100 CLSM, 1:500

**Table 2 biology-10-00402-t002:** TEM analysis of the GCX thickness in embedded THP-1 cells after HPF followed by the standard and optimized FS methods.

THP-1 Cells no.	Standard FS (+ OsO_4_)		Optimized FS (ø OsO_4_)	
	GCX (µm)	CB (µm)	GCX (µm)	CB (µm)
37	6.60 ± 0.30 *	13.71 ± 0.32 **	6.45 ± 0.26 *	13.11 ± 0.25 **

* mean length (±SEM) of the GCX of THP-1 cells; ** mean diameter (±SEM) of THP-1 cell body (CB).

## Data Availability

The data presented in this study are available on request from the corresponding authors.

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
