# Peer review of "Immuno-Electron and Confocal Laser Scanning Microscopy of the Glycocalyx"

_biology, 2021, doi:10.3390/biology10050402_

Round 1
Reviewer 1 Report
Based on previously established protocols, the authors developed a new protocol to for electron- and microscopic analysis of the GCX that preserves the morphology of GCX and warrants antigen accessibility. The experiments are well designed and executed.
I have a minor concern that need to be addressed: The immunogold labeling in Figure 7A can not be clearly visualized. The arrow is not clear and based on this, it is hard to tell whether the gold particle is associated with GCX or not. It would be good to enlarge Figure 7A to clearly indicate where are gold particles, and maybe show the particle in magenta as in Figure 4.
Author Response
Reviewer1
Based on previously established protocols, the authors developed a new protocol to for electron- and microscopic analysis of the GCX that preserves the morphology of GCX and warrants antigen accessibility. The experiments are well designed and executed.
I have a minor concern that need to be addressed: The immunogold labeling in Figure 7A can not be clearly visualized. The arrow is not clear and based on this, it is hard to tell whether the gold particle is associated with GCX or not. It would be good to enlarge Figure 7A to clearly indicate where are gold particles, and maybe show the particle in magenta as in Figure 4.
We followed the reviewers suggestion and modified figure 7.
We appreciate the positive feedback and are grateful for the suggestion to improve the manuscript.
Reviewer 2 Report
This paper describes the use of cryofixation, freeze-substitution, sample rehydration and immunogold labeling for EM and confocal microscopy of glycocalyx components in a monocytic cell line. Authors state several times they “developed” these techniques (e.g., lines 22, 25, 34-36 etc. & 477). This is, in my opinion not correct. There is no invention, justifying the term “develop(ment)”, since authors used slightly modified methods developed by other groups (van Doonselaar et al. 2007, Traffic 8, Ripper et al. 2008, Biol Cell 100, Hess et al. 2018, Traffic 19). Two of those publications are listed in the references, but are mentioned in the running text in a completely irrelevant context. In addition, I find several shortcomings in the presentation and discussion as listed in detail below. In the present state, the one-line summary of the paper could read: “We used cryofixation, freeze-substitution, sample rehydration and immunogold labeling for EM and confocal microscopy in a monocytic cell line and found x,w,z, etc. localizing at the pericellular glycocalyx structures….”.
Taken together, I am unable to recommend acceptance of this manuscript.
Points of concern and suggestions for possible improvement are:
Major:
Line 143: Please explain your decision to remove uranyl acetate for confocal microscopy. Ripper et al. 2008 showed successful immunofluorescence after freeze-substitution and rehydration with this reagent.
Line 158. Explain your decision not to postfix rehydrated samples with aqueous fixatives, although this step was found to be absolutely necessary by Ripper et al. 2008 for good structural preservation.
Line 256, 257 The best results of your freeze substitution test series differs only in one single minor point from the recipes and recommendations by vanDonselaar et al. 2007 and Ripper et al. 2008. You employ 0.2% glutaraldehyde, the others 0.5%. Does this make a significant difference in your experiments or not ? If so, please explain in detail; if not, do not highlight it. In any case, however, please mention the similarity with these two publications.
Line 444: “rehydration step”: This wording is really an inappropriate understatement: It neglects completely the fundamental principle and the difficulties encountered with rehydrating freeze substituted samples without introducing severe ultrastuctural damage. Establishing the right conditions (composition of freeze substitution media, rehydration media, choice of appropriate subzero temperatures at which specific rehydration steps are performed, postfixation...) took the Utrecht and Tübingen lab. years of hard experimental work. Therefore the superficial wording “rehydration step” is more than misleading for the non-expert reader. In a similar line I strongly recommend the authors to explain their choice to perform the entire rehydration procedure on ice (line 144), in contrast to all other colleagues starting at minus 35C (van Donselaar, Ripper…)
lines 482-485. All what the authors say here as part of their discussion has been explained and experimentally shown in detail in the publication by Hess et al 2018; the same is true for lines 490,491. Referring to this paper here in the text would be adequate and fair.
Minor:
Line 81 (and also on other places in the text, eg, line 431). “freeze substitution buffers”: the term buffer is wrong for a mixture made of organic solvents and fixatives; please replace by “freeze substitution media”, for example.
Line 147, 148: Please refer to your figure illustrating the workflow.
Line 189: gold particles cannot be crosslinked, please reword to “...crosslink gold conjugates”
Figures 2 and framed boxes in 4 lack scale bars, please add.
Line 356: For better illustration of the results obtained with the different techniques used a respective figure showing direct comparisons would be helpful and informative.
Line 412. Comment: The use of cell culture media for high-pressure freezing is not new, but may be essential with respect for glycocalyx preservation (see line 405).
finally: lines 69, 75 concerning Refs. 8,9. This is not direct criticism, just a point of interest to me: What do authors think about the appearance of a very loose, unstructured glycocalyx meshwork in Refs. 8,9, and their own work ,as compared to the completely different glycocalyx morphology shown by Reippert et al. 2018, J Histochem Cytochem 66 (glycocalyx of intestinal epithelium) or Brandt et al., 2021 Frontiers in Bioengineering and Biotechnology 8 (alveolar epithelium) from cryofixed, freeze-substitutted samples. Could you please include and comment this interesting aspect.
Reviewer 3 Report
The review entitled “Immunoelectron- and confocal laser scanning microscopy of the glycocalyx” introduces the method for microscopic analysis of the GLX. The detailed protocols for TEM and CLSM imaging are provided.
The manuscript has been prepared with good diligence, and leads from the basics of GLX structure and function to methods (and the methods limitations) for GLX imaging. Although the idea of such review seems to be very important and the growing number of papers in this field indicate the need for new methodologies for GLX imaging, the realization require some improvements.
- I recommend to add the list of abbreviations, that would make reading easier
- I have never met with “GCX” as the abbreviation of glycocalyx, rather “GLX” is used. Please, check which is commonly used and more correct
- I would like to know if the methodological protocols described in the paper are universal and can be applied to other systems. Have the authors tested the proposed methodology for other cells / tissues?
- Have the authors testes the negative control for TEM imaging? It means, the imaging of cells after enzymatic removal of GLX?
- How the Authors comment the thickness of GLX estimated from TEM (Table 2) and CLSM (data not provided) imaging, is it comparable?
- The degree Celsius symbol is in many places for editorial improvement (e.g. page 3, line 111)
In summary, the work is solid, well outlined and organized. Due to the all above-highlighted aspects, the manuscript in the present form can be recommended for publication in Biology after minor improvements.
Round 2
Reviewer 2 Report
Authors wrote a good rebuttal letter (which is appreciated) and exchanged singular misleading words in the manuscript.
However, a carefull revision of the manuscript is missing. None of the requested informations and explanations has been included in the text.
In addition, there are still some minor, but annoying mistakes that should be corrected. Together, I am sorry to insist on essential text revisions throughout the manuscript before supporting acceptance (note: the effort for those revisions is not extraordinary).
Major requests:
line 143: explain the reason for omitting UAc for the reader (not just the referee). What about a phrase like: "For CLSM we removed UAc from the FS-cocktail, SINCE THIS TOXIC AND RADIOACTIVE REAGENT IS NOT NEEDED FOR THIS PURPOSE." ?
Line 254: explain the cocktail of choice better for the reader. What about a phrase like: "We Tested FS methods avoidung osmium AND USING GENERALLY LOW CONCENTRATIONS OF FIXATIVES to ensure better preservation ...." ?
Minor corrections needed:
line 148: ...rehydrated in increasing concentrations of water/phosphate buffer IN ACETONE...
line 159: "...samples sections were incubated..." Meaning unclear: your specimens are cell suspensions.
line 269: "...GCX.. regularly arranged..". Which kind of regularity ? There is no oder evident, as in periodic collagen, Fibonacci-sequences of leave/flower arrangement, or stratified tissue.
line 274-275: "...because the membranes are impermeable to UAc." This is wrong, since UAc preferentially bind to phospholipids (see G. Griffiths: Fine structure Immunocytochemistry 1993, M. Hayat - whatever edition). Just remove this part of the sentence.
line 322-323: again: "buffer" for FS....
line 338-339: no criticism, just a suggestion to make your statement stronger: Labeling after chemical fixation was not only strictly confined to the membrane, but was also WEAK; the signal after HPF, FS, RH was not only diffuse but also much STRONGER. Do emphasize this important point.
line 365: "..ordered mesh-like arrangements": same as line 269: just remove "ordered".
Line 414-416: "This allowed...culture medium." This sentence is highly misleading. There is no direct, causal link between FCS and centrifugation. I suppose the sentence should read: "This allowed to concentrate the cells by sedimentation IN ORIGINAL MEDIUM WITHOUT MEDIA CHANGES AND ADDITIONAL CENTRIFUGATION."
line 447. Remove: "AND OBVIOUSLY EFFICIENTLY RESTORED THE EPITOPE IMMUNOREACTIVITY AFTER FS". This part is wrong, since antigenicity was not detroyed or masked during FS, therefore no need to "RESTORE" it.
line 465-466. reword: "With slight modifications in the FS procedure, we were able to translate..." This wording gives no logical argument. The modifications were not obligatory for successfull CLKM. I suggest simply to write: "...in the FS PROCEDURE, WE TRANSLATED OUR PROTOCOL...".
Author Response
Please see attachement

Round 3
Reviewer 2 Report
Now it's well done. Fine. The present form of your contribution is clear and helps the interested reader to understand your concept. And this is a good basis for further scientific progress.